# A New Image Encryption Algorithm Based on Composite Chaos and Hyperchaos Combined with DNA Coding

**DOI:** 10.3390/e22020171

**Published:** 2020-02-02

**Authors:** Yujie Wan, Shuangquan Gu, Baoxiang Du

**Affiliations:** 1Electronic Engineering College, Heilongjiang University, Harbin 150080, China; 2181237@s.hlju.edu.cn (Y.W.); 2181235@s.hlju.edu.cn (S.G.); 2Kunpad Communications(KunShan) Co., Ltd., Kunshan 215300, China

**Keywords:** chaotic systems, image encryption, DNA coding, security analysis

## Abstract

In order to obtain chaos with a wider chaotic scope and better chaotic behavior, this paper combines the several existing one-dimensional chaos and forms a new one-dimensional chaotic map by using a modular operation which is named by LLS system and abbreviated as LLSS. To get a better encryption effect, a new image encryption method based on double chaos and DNA coding technology is proposed in this paper. A new one-dimensional chaotic map is combined with a hyperchaotic Qi system to encrypt by using DNA coding. The first stage involves three rounds of scrambling; a diffusion algorithm is applied to the plaintext image, and then the intermediate ciphertext image is partitioned. The final encrypted image is formed by using DNA operation. Experimental simulation and security analysis show that this algorithm increases the key space, has high sensitivity, and can resist several common attacks. At the same time, the algorithm in this paper can reduce the correlation between adjacent pixels, making it close to 0, and increase the information entropy, making it close to the ideal value and achieving a good encryption effect.

## 1. Introduction

With the rapid development of the Internet, more and more multimedia image information is transmitted online. Images are widely used because of their vivid and intuitive characteristics. People can easily access other people’s information through the Internet with the help of an ordinary computer and network cable. Therefore, the question of how to transfer the information safely and ensure its security has become an urgent problem to be solved. Image encryption is the primary solution. Due to high redundancy and correlation between image pixels, large amounts of data, and fidelity, traditional text encryption technology cannot meet the needs of image encryption [1]. Therefore, the development of secure and effective image encryption algorithms is still the focus of the communication field [2].

Due to its high sensitivity to initial values and system parameters, excellent ergodicity, and good pseudo-randomicity, chaotic systems have become the primary choice of cryptographic systems [3,4]. Therefore, many image encryption schemes based on chaos have been proposed [5,6]. Among them, chaotic image encryption methods are divided into one-dimensional chaotic and multidimensional chaotic encryption methods. A one-dimensional chaotic system has a simple structure which is easy to implement. However, they also have some problems: the scope of chaotic behavior is small, and the Lyapunov index is low [7]. Some improved encryption schemes for one-dimensional chaotic maps have been proposed. Wu et al. improved the existing one-dimensional chaos and proposed a new image encryption method [8]. A new method was proposed by Chao et al. who took the output of tent mapping as the input of Chebyshev mapping, and then applied perturbations to generate excellent pseudo-random chaotic sequences for encryption [9]. Hua et al. proposed to combine two one-dimensional chaotic systems in parallel to form a new one-dimensional chaotic system through cosine transform to encrypt the image [10], which increased the scope of system chaotic mapping. C P et al. defined a new one-dimensional chaotic map with the difference of two chaotic output sequences [11]. These methods expand the scope of chaotic mapping and improve chaotic properties to some extent, but the system parameters are still limited. On the other hand, the multi-dimensional chaotic phase space is complex, the system parameters have more flexibility, and the dynamic behavior is difficult to predict. In particular, the hyperchaotic system has two or more positive Lyapunov exponents, and the characteristics of the chaos are better for this system. A multi-dimensional chaotic system can produce multiple chaotic sequences at the same time, which can be used in image scrambling and diffusion, respectively, with high security. Sun adopts a 5-D hyperchaotic system to generate pseudo-random sequences and decompose permutation images, which can resist statistical attacks and differential attacks and is suitable for practical application [12].

Since DNA molecules can be processed in parallel on a large scale, with huge storage and ultra-low power consumption, many image encryption methods are proposed by many researchers who combine chaotic mapping and DNA coding technology. In 1994, Aldeman proposed DNA computing for the first time, ushering in a new era of information processing [13]. In 2002, Gehani et al. proposed to encrypt images one by one with DNA strings [14]. In 2012, an image encryption method based on piecewise linear mapping of DNA and PWLCM was proposed, which increased the key space [15]. However, these encryption methods cannot resist selective plaintext attacks and known plaintext attacks. In 2019, Zhang et al. proposed a new image encryption method based on quantum chaos and DNA coding, which has high security and can resist brute force attacks and statistical attacks [16]. In 2019, a color image encryption algorithm based on dynamic DNA encryption and chaos was proposed, using the hash function and external parameters to calculate its initial value, which can effectively resist the selected plaintext attack with better security [17]. Guan et al. proposed a digital image encryption algorithm based on DNA and frequency domain hyperchaos, which improved security against differential attacks [18]. Yang et al. proposed an image compression and encryption scheme based on fractional-order hyperchaotic system combining 2D compressed sensing and DNA coding. The fractional order and initial value of the fractional hyperchaos system are used as the key of the encryption scheme, which greatly expands the key space and has a strong ability to resist multiple attacks [19].

In order to provide a better encryption effect, a new image encryption scheme based on double chaos (one-dimensional composite chaos and hyperchaos) and DNA coding technology is proposed. This algorithm has the following advantages: (1) First, Fibonacci transformation and diffusion operation of modularization are performed on the plaintext image, and the pixel position and value of the plaintext image are fully changed to reduce the image correlation. (2) The first-round scrambling-diffusion operation is repeated three times, so that the value of each encrypted pixel is affected by the previous one, which increases the sensitivity to its clear text. (3) A new one-dimensional complex chaos is proposed, which has no period window within the chaos scope, that is to say it is a full map, and is larger than the corresponding one-dimensional chaotic Lyapunov exponents. Combining the new chaotic sequence with DNA technology, the secondary encryption extends the complexity and improves its security. (4) Taking the pixel value of the plaintext image as the initial value of the chaotic system can resist the plaintext attack and increase the key space. In this paper, key space, statistical analysis, differential attack, and anti-noise attack are analyzed. Experimental results and security analysis also confirm that the algorithm proposed in this paper increases key space, has high sensitivity, can resist multiple attacks, and can effectively protect the security of image information.

The rest of this paper is arranged as follows. The second section mainly introduces the theoretical knowledge required in this paper, such as typical chaotic systems, newly constructed LLS system, and DNA coding technology. The third section introduces in detail the image encryption scheme based on double chaos and DNA coding technology. The fourth section is the experimental simulation and security analysis. Finally, the fifth section draws the conclusion of this paper.

## 2. The Basic Principle

### 2.1. One-Dimensional Chaotic Mapping

#### 2.1.1. Logistic Chaotic Mapping

Logistic chaotic mapping is a classical one-dimensional chaotic mapping with a simple structure and few control parameters, which is convenient for implementation and generalization involving other chaos [20]. The expression of Logistic chaotic mapping is shown as Formula (1):(1)$$x_{n+1}=\mu x_{n}(1-x_{n}),\;n=0,1,2,3\cdots$$
where *μ* is the system control parameter, and $x_{0}$ is the initial value of the system $0<x_{0}<1$. The bifurcation diagram and lyapunov exponent of logistic chaotic mapping are shown in Figure 1a and Figure 2a. It can be seen that with the increase of *μ* and the number of bifurcations of the system, when *μ* varies from 3.5699456 to 4, the system enters a chaotic state.

#### 2.1.2. Sine Chaotic Mapping

Sine chaotic mapping is a mapping derived from the Sine function, which can convert the input angle in the range from 0 to 1/π to the output angle in a certain range [21]. The expression of Sine chaotic mapping is shown as Formula (2):(2)$$x_{n+1}=rsin(\pi x_{n})/4,$$
where $x_{n}$ is the input and r is the system control parameter. The bifurcation diagram of Sine’s chaotic mapping and lyapunov exponent are shown as Figure 1b and Figure 2b.

### 2.2. LLSS Chaotic Mapping

A new one-dimensional chaotic system can be obtained by using the existing one-dimensional chaos as a seed map. In this paper, two logistic maps and Sine map were connected in parallel, and then a mod operation was used to form a new one-dimensional chaos algorithm named LLSS. The structure is shown as Figure 3. The system expression is defined by Formula (3):(3)$$x_{n+1}=mod(2ax\left(n\right)\left(1-x\left(n\right)+\left(8-2a\right)\sin\left(\pi x\left(n\right)\right)/4,1\right).$$

The bifurcation diagram and Lyapunov exponent of LLSS are shown in Figure 1c and Figure 2c. As can be seen from the figure, the LLSS is fully mapped within the range of [0,4] and has no period window. Compared with the classical one-dimensional chaotic map, the Lyapunov exponent also increases.

### 2.3. Qi Hyperchaotic System

In 2005, Qi et al. discovered and named a new chaos algorithm called the Qi chaotic system [22]. On the basis of the experience of increasing dimensions to obtain hyperchaos, Qi et al. further proposed the Qi hyperchaos system. In comparison, the dynamic characteristics are more complex and the motion trajectory traversal range in phase space is larger [23]. The Qi hyperchaotic system is a four-dimensional hyperchaotic system. The dynamic equation is shown as Formula (4) as follows:(4)$$\{\begin{array}{c} \dot{x}=a\left(y-x\right)+yzw\\ \dot{y}=b\left(x+y\right)-xzw\\ \dot{z}=-cz+exyw\\ \dot{w}=-dw+xyz \end{array}$$

When the system parameters *a* = 50, *b* = 4, *c* = 13, *d* = 20, *e* = 4, the system is in a hyperchaotic state. When the initial value [1; 2; 3; 4] is selected, its attractor phase diagram develops as shown in Figure 4.

### 2.4. DNA Coding Technique

A DNA sequence is a string of molecules that represent the genetic information carried. The sequence consists of four deoxyribonucleic acids, which are A(adenine), T(thymine), C(cytosine), and G(guanine) [24]. A and T as well as C and G are complementary pairs. When applying DNA sequences to binary Numbers, 0 and 1 are complementary. Four deoxyribonucleic acids are represented by two binary Numbers, so 00 and 11 are complementary, and 01 and 10 are also complementary. There are eight combinations satisfying the principle of base complementary pairing, that is, there are eight combinations of coding rules [25].

Plaintext can be thought of as a matrix with a pixel value from 0 to 255, and each plaintext pixel can be represented by a DNA sequence with a length of 4. For example, this information with a pixel value of 182 is converted into a binary sequence [10110110], which is encoded according to coding rule 1 in Table 1. The binary sequence obtained is [10111001], and the corresponding DNA sequence is CTGC. According to coding rule 2, the binary sequence obtained is [01111001]. DNA operations include XOR, addition, and subtraction, represented by a ternary number, where 0 represents DNA XOR, 1 represents DNA addition, and 2 represents DNA subtraction. These three operation rules between DNA sequences are set as shown in Table 2.

## 3. Proposed Encryption Algorithm

The flow chart of the proposed encryption scheme is shown in Figure 5. Suppose that the size of the original image I is M × N, and the encryption process is as follows:

Step 1: read in the original image I and use the Fibonacci transform to produce scrambled image F.

Definition: Fibonacci is a scrambling algorithm based on two-dimensional chaotic mapping, which is a nonlinear transformation in modular form and reduces the correlation by changing the position relation of image pixels. Its definition is shown in Formula (5):(5)$$\left(\begin{array}{c} x^{\prime }\\ y^{\prime } \end{array}\right)=\left(\begin{array}{c} 1\;\;\;\;1\\ 1\;\;\;\;0 \end{array}\right)\left(\begin{array}{c} x\\ y \end{array}\right)\;mod\;N.$$

Step 2: The scrambled image F is diffused with the algorithm of adding and taking modules to obtain the diffusion image K. The main Formula is shown in Formula (6). This diffusion operation can make the scrambled image fully diffuse into the ciphertext,
(6)$$K_{i}=(K_{i-1}+H_{i}+F_{i})\;mod\;256$$
where $K_{i}$ is the diffused image, $F_{i}$ is the scrambled image, and $H_{i}$ is the password pixel.

Step 3: Repeat step 1 and step 2 for the three times to fully obtain the middle ciphertext *M*.

Step 4: Generate an M × M random matrix using LLSS chaotic mapping denoted as R. Given the initial value and system parameters of LLSS, the chaotic sequence of LLSS is generated by iterating SUM + 999 times, and the first 1000 points are removed to obtain the sequence P, which is transformed into an integer from 0 to 255, and then transformed into a random matrix R of M rows and N columns.

Step 5: construct a control sequence with a hyperchaotic Qi system
(1)In order to resist the selective plaintext attack, the relationship between the initial value of the system and the plaintext is established, and the initial value of the hyperchaotic system $X_{0}$, $Y_{0}$, $Z_{0}$ and $W_{0}$ is obtained according to the Formula (7) to (10).(2)In order to obtain better randomness, the first 1500 iterations is removed and four hyperchaotic sequences X, Y, Z and W are generated. To reconstruct the sequence, X and Y determine the encoding mode of DNA, Z determines the operation of DNA, and W represents the decoding mode of DNA.
(7)$$X_{0}=sum\left(sum\left(bitand\left(I,3\right)\right)\right)/\left(3*SUM\right),$$
(8)$$Y_{0}=sum\left(sum\left(bitand\left(I,12\right)/4\right)\right)/\left(3*SUM\right),$$
(9)$$Z_{0}=sum\left(sum\left(bitand\left(I,48\right)/16\right)\right)/\left(3*SUM\right),$$
(10)$$\;\;W_{0}=sum\left(sum\left(bitand\left(I,192\right)/64\right)\right)/\left(3*SUM\right),$$

Step 6: The random matrix R and the middle image M are preprocessed and divided into four blocks. The middle image M is encoded according to the sequence number corresponding to X to get D1, and the random matrix R is encoded according to the sequence number corresponding to Y to get D2. Then the above two encoded blocks are calculated according to Z. Finally, the results of the operation are calculated with the results of the previous one again. Combine the split blocks to get the final encrypted image E.

Decryption is the reverse operation of encryption. Decryption is mainly divided into three modules: DNA decoding and operation, inverse diffusion operation, and inverse operation of Fibonacci transformation. These modules are shown in the lower part of Figure 5.

## 4. Simulation Results and Security Analysis

The five images size of are used as the test images 256 × 256 including Lena, Couple, Cameraman, Baboon, and Lake. Simultaneous, the MatlabR2015a is used as the platform. The original image, the encrypted image, and the corresponding decrypted image are shown in Figure 6. It can be seen from the comparison diagram that the encrypted image is a snowflake, in which there is no information of the original image, and the original image can also be decrypted from the encrypted image, indicating that the algorithm proposed in this paper has a good encryption effect. In this section, the proposed algorithm is analyzed for security.

### 4.1. Key Analysis

#### 4.1.1. Key Space

A good encryption algorithm should have enough key space to resist exhaustive attacks. The key of the proposed algorithm consists of a total of seven keys: $x_{0}$, $y_{0}$, $z_{0}$, $w_{0}$, $H_{0},x_{01}$, and $\mu_{0}$. According to the international standard IEEE 754, the index portion is expressed as a positive value to simplify the comparison. The significant digit of a double-precision floating-point type is 52 bits, the size of the key space of the control parameter will be greater than $2^{52\times 7}=2^{364}>2^{128}$. The results show that it is almost impossible to attack the algorithm correctly by brute force, so the encryption algorithm can resist brute force attacks.

#### 4.1.2. Key Sensitivity

A small change in the decryption key makes a huge difference to the result, and the original image will not be decrypted correctly, indicating that the algorithm gas has a high sensitivity. First, set the initial values of the Qi hyperchaos system: $x_{0}$ = 0.5001, $y_{0}$ = 0.5130, $z_{0}$ = 0.5170, $w_{0}$ = 0.3237; and the initial values of the LLSS system: $x_{01}$ = 0.3711, $\mu_{0}$ = 3.9990. Then, make a tiny change to the encryption key, select one of the key parameters, and add $10^{-10}$ so that the results can be compared as shown in Figure 7. It can be seen that only a slight change can have a huge effect. And the decryption diagram is completely different from the original image. Therefore, it can be concluded that it is impossible to decrypt by completely guessing the encryption key.

### 4.2. Statistic Analysis

#### 4.2.1. Gray Histogram

Gray histogram is more intuitive, and the visibility is good. It can be intuitively seen from the figure that the frequency or probability of occurrence of the gray value. The more balanced the histogram, the better the encryption effect [26]. The comparison results are shown in Figure 8. The gray level histogram represents each gray level and the number of times that gray level occurs. The x-axis represents grayscale values of 0 to 255, and the y-axis represents the number of pixels in the corresponding grayscale in the figure. As can be seen from the figure, the histogram of the original image fluctuates greatly and is not uniform; Ciphertext images are roughly evenly distributed. The results show that the attacker cannot get information about the original image from the ciphertext, which indicates that the algorithm proposed in this paper has a good encryption effect.

#### 4.2.2. Correlation Analysis of Adjacent Pixels

Two thousand pairs of adjacent pixel values are randomly selected from the horizontal, vertical and diagonal directions of plaintext and ciphertext images. The following Formulas (11) to (14) are used to calculate the correlation coefficient of two adjacent pixel values:(11)$$\rho_{xy}=\frac{cov\left(x,y\right)}{\sqrt{D\left(x\right)}\sqrt{D\left(y\right)}},$$
(12)$$E\left(x\right)=\frac{1}{N}\sum_{i=1}^{N}x_{i},$$
(13)$$D\left(x\right)=\frac{1}{N}\sum_{i=1}^{N}{\left(x_{i}-E\left(x\right)\right)}^{2},$$
(14)$$cov\left(x,y\right)=\frac{1}{N}\sum_{i=1}^{N}\left(x_{i}-E\left(x\right)\right)\left(y_{i}-E\left(y\right)\right),$$
where *x*, *y* is the gray value of two adjacent pixels in the image, N is the total pixel value selected from the image, *E*(*x*) and *E*(*y*) are the mean value, *D*(*x*) and *D*(*y*) are the variance. The smaller the absolute value of the correlation coefficient is, the lower the correlation is. The correlation coefficient of plaintext and ciphertext is shown in Table 3. It can be seen from Table 3 that the absolute value of plaintext image correlation is close to 1, and the absolute value of ciphertext correlation is close to 0, which indicates that the image correlation after encryption is destroyed. The correlation diagram is shown in Figure 9, from which it can be seen that the pixels of the plaintext image are highly concentrated and distributed near the corners, while the pixels of the ciphertext image are evenly distributed.

#### 4.2.3. Information Entropy

The information entropy of the image is considered from the statistical characteristics and represents the overall characteristics of the image in the mean sense. It reflects the average amount of information in the image. The following Formula (15) is used to calculate the information entropy of the image:(15)$$H\left(x\right)=\sum_{i=0}^{2^{n}-1}p\left(m_{i}\right)log_{2}\frac{1}{p(m_{i})},$$
where $p(m_{i})$ represents the probability of signal m. For a 256 × 256 image, the ideal value of entropy is equal to 8, which means the image is uniform. The closer it gets to 8, the harder the cryptosystem leaves some information available. When the probability of each gray value is basically equal, the entropy reaches the maximum value. Table 4 is the information entropy of the algorithm proposed in this paper. It can be seen from Table 4 that the information entropy of this paper is close to 8, which indicates that the probability of accidental information leakage is very small.

### 4.3. Differential Attack

The difference between plaintext and ciphertext can be expressed by NPCR (the number of pixels change rate) and UACI (the number average changing intensity), where NPCR represents the ratio of different gray values of different ciphertext images at the same position, while UACI represents the average change density of different ciphertext images. UACI and NPCR can be used to test the ability of encryption algorithms to resist differential attacks. The Formulas (16) to (18) are to calculate NPCR and UACI.
(16)$$NPCR=\frac{\sum_{i,j}D\left(i,j\right)}{M\times N}\times 100\%,$$
(17)$$UACI=\frac{1}{M\times N}\times \frac{\sum_{i,j}\left|C_{1}\left(i,j\right)-C_{2}\left(i,j\right)\right|}{L}\times 100\%,$$
(18)$$D\left(i,j\right)=\{\begin{array}{cc} 0, & C_{1}\left(i,j\right)=C_{2}\left(i,j\right)\\ 1, & otherwise \end{array},$$
where $C_{1}\left(i,j\right)$ and $C_{2}\left(i,j\right)$ represent the ciphertext image corresponding to two plaintext images with only one pixel difference. For a 256-level image, the ideal values of UACI and NPCR are 33.4635% and 99.6094%. The test results are shown in Table 5. It can be seen from the table that the average UACI is 99.6130% and NPCR is 33.5211%, which is very close to the ideal value.

### 4.4. Anti-Noise Ability

In order to test the anti-noise ability of the algorithm, add a different intensity of Salt and Pepper noise and Gaussian noise to the ciphertext image and decrypt it. Then use the peak signal to noise ratio (PSNR) to assess it, which is the most widely used image perception quality evaluation method, and defined by the mean square error (MSE):(19)$$MSE=-\frac{1}{m\times n}\sum_{i=1}^{m}\sum_{j=1}^{n}{\left[I\left(i,j\right)-D\left(i,j\right)\right]}^{2},$$
(20)$$PSNR=10\mathrm{lg}\left(\frac{255^{2}}{MSE}\right),$$
where *I* is the original image and *D* is the decrypted image. The test results are shown in Table 6. First increase the noise of the density of 0.001, 0.005 and 0.01 to the cipher images. The noised cipher images are shown in the first column of Table 6, and then they can be decrypted. The decrypted images are shown in the third column of Figure 6. The corresponding PSNR is shown in the fourth column. It can be seen from the figure that in the case of noise, the algorithm in this paper can decrypt the noised cipher images and obtain the original image information. Even if the noise intensity reaches 0.01, the decrypted image can still be visually recognized. It can be seen that the encryption scheme can effectively resist a certain degree of noise attack.

### 4.5. Anti-Cropping Ability

To test the ability of the proposed algorithm to resist clipping attacks, set the gray values of some pixels of the encrypted image to 0, and then decrypt it with the correct key. As shown in Figure 10, it can be seen that after cutting off a pixel block, the original image can still be decrypted to a certain extent, indicating that the algorithm proposed in this paper has a certain degree of anti-cropping ability.

### 4.6. Chosen-Plaintext Attack

In cryptanalysis, there are four typical attacks: ciphertext-only attack, known-plaintext attack, chosen-plaintext attack, and chosen-ciphertext attack. If it can resist a chosen-ciphertext attack, it has enough security to resist other attacks. In this paper, two kinds of images, all black and all white, are used for testing. The encryption diagram and its histogram are shown in Figure 11. At the same time, the correlation between information entropy and adjacent pixels can be analyzed, as shown in Table 7.

### 4.7. Comparative Analysis with Other Literatures

The algorithm proposed in this paper is compared with other literatures in terms of key space, information entropy and differential attack. The results are shown in Table 8. It can be seen from the table that the algorithm proposed in this paper is close to the ideal value, and better than the algorithms discussed in other literatures in three ways, indicating that this algorithm has a good encryption effect.

The algorithm proposed in this paper is compared with other literatures on related rows of adjacent pixels. The results are shown in Table 9. As can be seen from the table, the algorithm proposed in this paper reduces the pixel correlation from the three directions of horizontal, vertical, and diagonal, so that its absolute value is close to 0. Compared with other algorithms, the reduction effect of this algorithm is better.

### 4.8. Structural Similarity Index (SSIM)

SSIM is a measure of the similarity of two images. If the two images are before encryption and after decryption, then SSIM can be used to evaluate the quality of the encrypted image. The value is from 0 to 1. The larger the value, the smaller the image distortion. Calculated as follows:(21)$$\mu_{X}=\frac{1}{m\times n}\sum_{i=1}^{m}\sum_{j=1}^{n}X\left(i,j\right),$$
(22)$$\sigma_{X}=\frac{1}{m\times n-1}\sum_{i=1}^{m}\sum_{j=1}^{n}{\left(X\left(i,j\right)-\mu_{X}\right)}^{2}{)}^{1/2},$$
(23)$$\sigma_{XY}=\frac{1}{m\times n-1}\sum_{i=1}^{m}\sum_{j=1}^{n}\left(X\left(i,j\right)-\mu_{X}\right)(Y\left(i,j\right)-\mu_{Y}),$$
(24)$$SSIM=\frac{\left(2\mu_{X}\mu_{Y}+C_{1}\right)\left(2\sigma_{XY}+C_{2}\right)}{\left(\mu_{X}{}^{2}+\mu_{Y}{}^{2}+C_{1}\right)\left(\sigma_{X}{}^{2}+\sigma_{Y}{}^{2}+C_{2}\right)},$$
where $C_{1}={\left(0.01\times 255\right)}^{2}$, $C_{2}={\left(0.03\times 255\right)}^{2}$. Calculate the SSIM value is 0.81085 according to the formula. It can be seen that it is within the range and the value is relatively high. This shows that the algorithm has less distortion.

### 4.9. Computational Complexity Analysis

The image encryption algorithm was implemented by Matlab on a personal computer with an Intel i5-4210U processor and 4.00G RAM. It takes time to record the encryption and decryption of different image sizes. The results are shown in Figure 12.

## 5. Discussion

This paper proposes a new one-dimensional chaos, which is formed by parallel processing of Logistic and Sine chaos as seed maps and through modulo operation. The new chaos has the advantages of a simple one-dimensional chaotic structure, being easy to implement and full mapping in the chaos range. The algorithm in this paper is based on the combination of the double chaos, this new one-dimensional chaotic, and hyperchaos Qi, and uses DNA coding technology to achieve image encryption. In the fourth part of the experimental simulation and performance analysis, we can see that the algorithm proposed in this paper can increase the key space, have high sensitivity to the key, reduce the degree of correlation of the original image, and resist the advantages of multiple attacks. However, the efficiency of the algorithm discussed in this paper is not high, and the degree of anti-attack needs to be improved. This will be progressed in future research.

## 6. Conclusions

In this paper, a new image encryption scheme based on composite chaos and Qi hyperchaos combined with DNA coding is proposed. In this scheme, Fibonacci transformation and diffusion algorithm of adding modules are used for initial encryption. Then the intermediate ciphertext and the new compound chaos are calculated by DNA to form the final ciphertext. In order to resist chosen-plaintext attack, the algorithm takes the sum of original image pixels as the initial value of a chaotic sequence. Experimental simulation shows that this scheme can increase the key space and resist many common attacks. However, the efficiency of the scheme is not high, so the main work in the future will be to improve the efficiency of the algorithm.

## Figures and Tables

**Figure 1 entropy-22-00171-f001:**
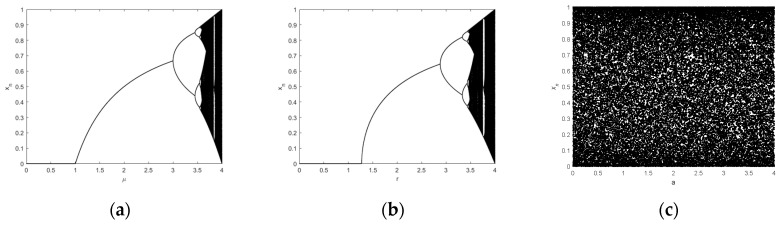
The Bifurcation diagrams of the (**a**) Logistic map, (**b**) Sine map, (**c**) LLSS map.

**Figure 2 entropy-22-00171-f002:**
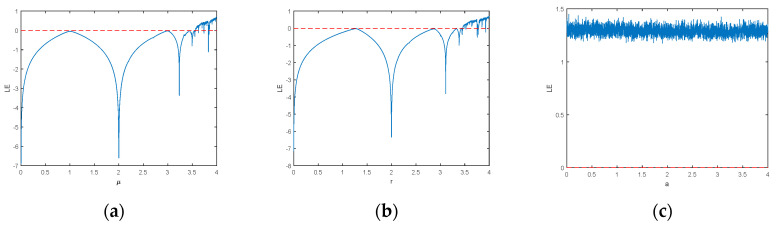
The Lyapunov Exponent of the (**a**) Logistic map, (**b**) Sine map, (**c**) LLSS map.

**Figure 3 entropy-22-00171-f003:**
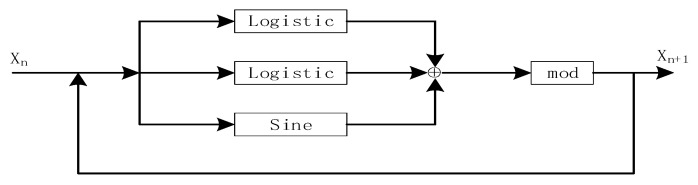
The new chaotic system of the LLS map.

**Figure 4 entropy-22-00171-f004:**
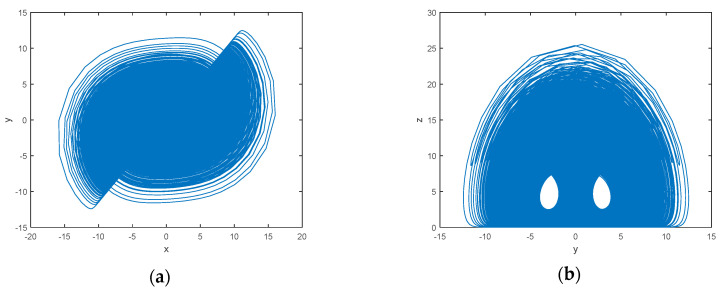
Qi Hyper-chaotic attractor: (**a**) (x-y) plane; (**b**) (y-z) plane; (**c**) (x-w) plane; (**d**) (x-y-z) plane.

**Figure 5 entropy-22-00171-f005:**
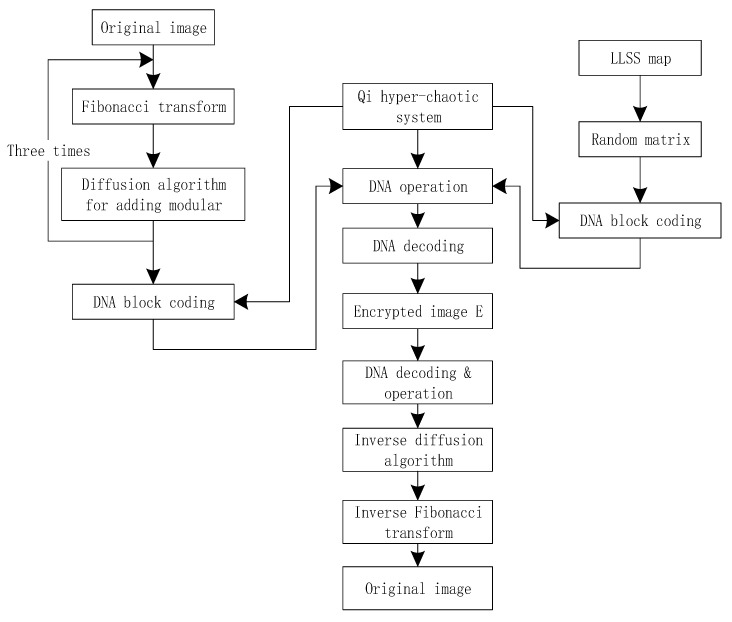
The flow chart of the proposed image encryption algorithm.

**Figure 6 entropy-22-00171-f006:**
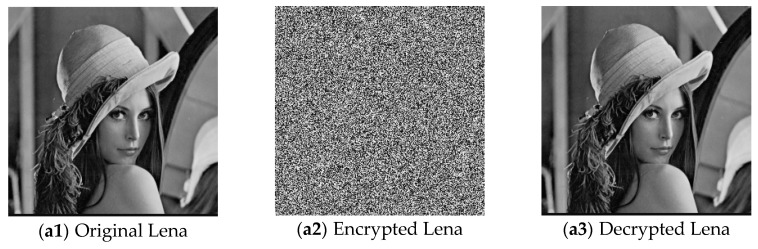
Original (**a1**)–(**e1**), encrypted (**a2**)–(**e2**), and decrypted of test image (**a3**)–(**e3**).

**Figure 7 entropy-22-00171-f007:**
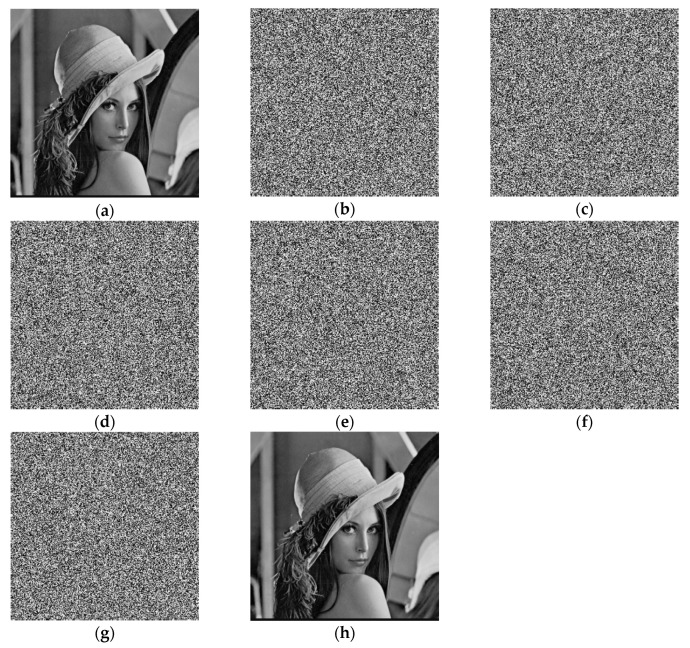
Key sensitivity: (**a**) Lena (**b**) $x_{0}=0.5001+10^{-10}$, (**c**) $y_{0}=0.5130+10^{-10},$ (**d**) $z_{0}=0.5170+10^{-10},$ (**e**) $w_{0}=0.3237+10^{-10},$ (**f**) $x_{01}=0.3711+10^{-10},$ (**g**) $x_{0}=3.9990+10^{-10}$, (**h**) corrected decrypted image.

**Figure 8 entropy-22-00171-f008:**
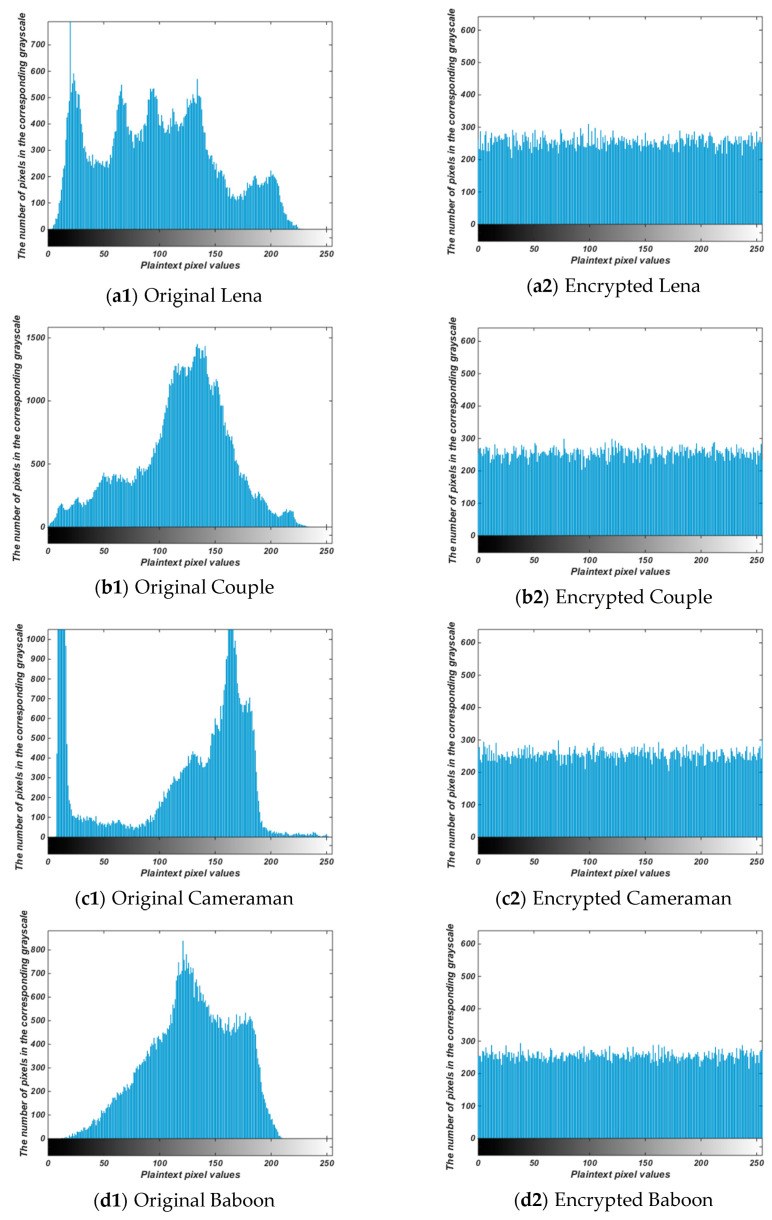
Gray Histogram of original (**a1**)–(**e1**). Gray Histogram of decrypted image (**a2**)–(**e2**).

**Figure 9 entropy-22-00171-f009:**
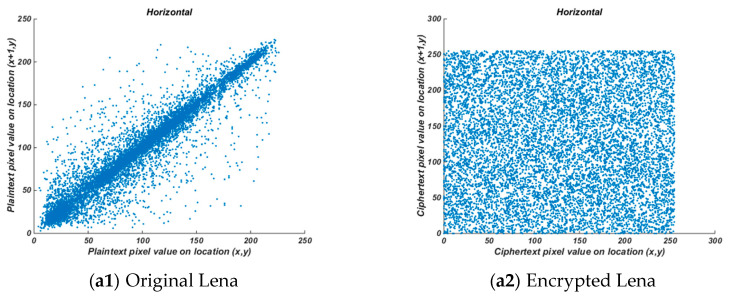
Horizontal correlation of adjacent pixels of original (**a1**)–(**e1**), encrypted image (**a2**)–(**e2**).

**Figure 10 entropy-22-00171-f010:**
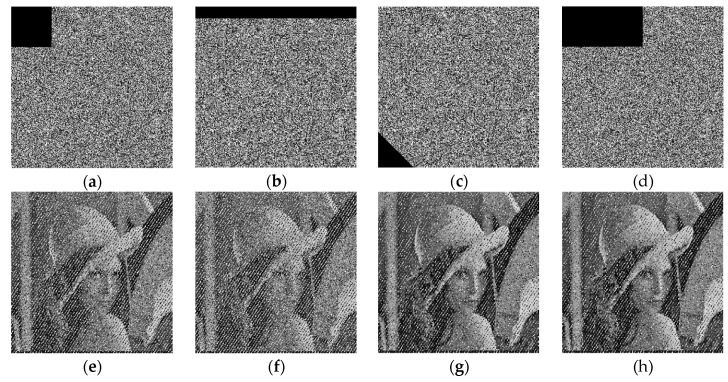
Cropping attacks with different areas. (**a**–**d**) Partially cut the encrypted image, (**e**–**h**) decrypted images.

**Figure 11 entropy-22-00171-f011:**
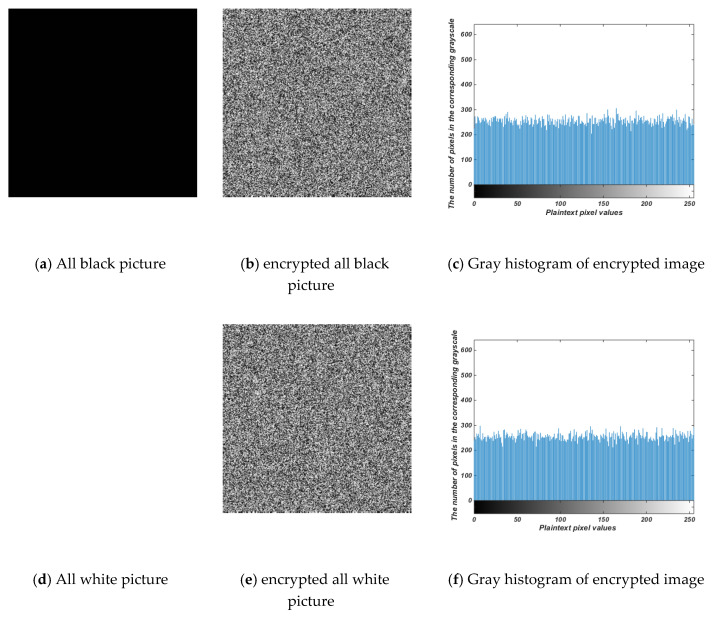
Test results with all black and all white.

**Figure 12 entropy-22-00171-f012:**
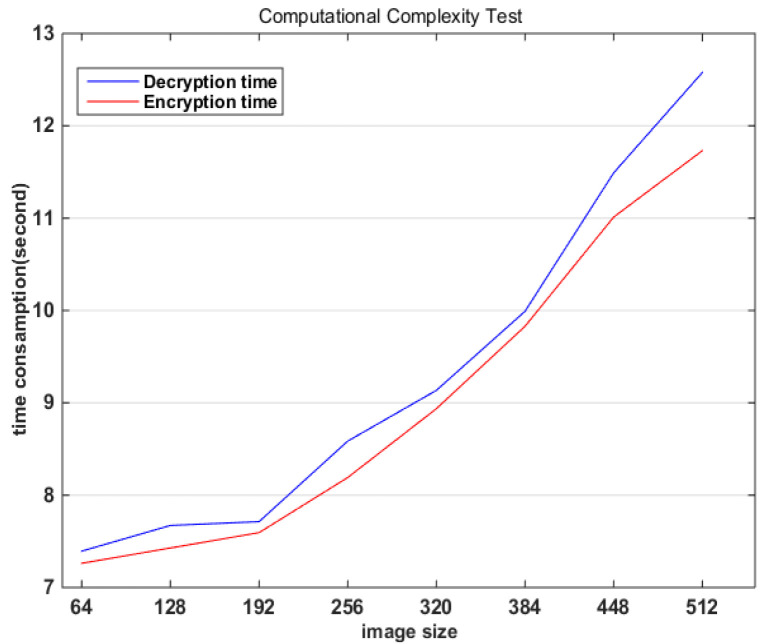
Image encryption algorithm computational complexity test.

**Table 1 entropy-22-00171-t001:** DNA coding rules.

Title 1	1	2	3	4	5	6	7	8
A	00	00	01	01	10	10	11	11
T	11	00	10	10	01	01	00	00
C	01	10	00	11	00	11	01	10
G	10	01	11	00	11	00	10	01

**Table 2 entropy-22-00171-t002:** DNA XOR, Addition and Subtraction.

XOR	A	G	C	T	+	A	G	C	T	-	C	A	T	G
A	A	G	C	T	A	A	G	C	T	C	C	A	T	G
G	G	A	T	C	G	G	C	T	A	A	G	C	A	T
C	C	T	A	G	C	C	T	A	G	T	T	G	C	A
T	T	C	G	A	T	T	A	G	C	G	A	T	G	C

**Table 3 entropy-22-00171-t003:** Correlation coefficients of adjacent pixels for the test images.

Image	Scheme	Horizontal	Vertical	Diagonal
Lena	Original image Cipher image	0.93767 0.0020306	0.97178 0.010543	0.9104 0.0019857
Couple	Original image Cipher image	0.9485 0.0031994	0.93625 0.0044791	0.89823 −0.000148
Cameraman	Original image Cipher image	0.92127 0.0026387	0.9633 0.010641	0.89823 −0.000148
Baboon	Original image Cipher image	0.90552 −0.014249	0.9228 0.0073645	0.8557 0.0068203
Lake	Original image Cipher image	0.93051 0.0012594	0.95735 −0.0014642	0.89664 0.0020329

**Table 4 entropy-22-00171-t004:** Information entropy.

Image	Lena	Couple	Cameraman	Baboon	Lake
Original image	7.5534	7.4601	7.0097	7.3649	7.5314
Cipher image	7.9974	7.9971	7.9970	7.9968	7.9973

**Table 5 entropy-22-00171-t005:** UACI and NPCR.

Image	Lena	Couple	Cameraman	Baboon	Lake	Average
NPCR (%)	99.5987	99.6276	99.6002	99.6170	99.6216	99.6130
UACI (%)	33.5267	33.5208	33.3921	33.6318	33.5344	33.5211

**Table 6 entropy-22-00171-t006:** PSNR with different noises and intensities.

Noise	Noisy encrypted images	Noise intensities	Decrypted images	PSNR(dB)
Salt and Pepper noise	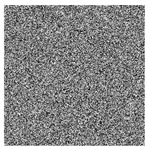	0.001	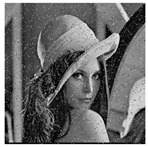	41.7268
	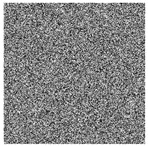	0.005	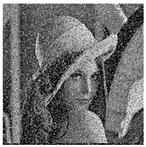	34.7189
	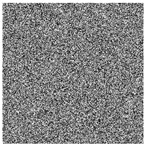	0.01	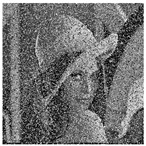	33.4257
	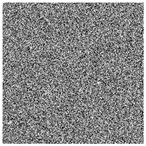	0.001	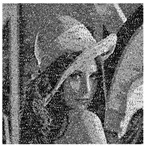	35.2165
Gaussian	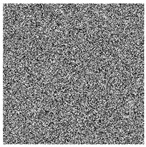	0.005	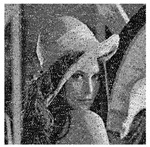	33.8192
	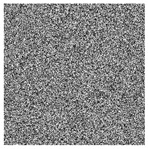	0.01	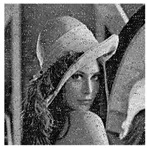	32.483

**Table 7 entropy-22-00171-t007:** Information entropy and correlation coefficients of the test images.

	Correlation Coefficients			
	Entropy	Horizontal	Vertical	Diagonal
All black	0	—	—	—
Cipher with all black	7.9972	−0.0036	0.0261	0.0033
All white	0	—	—	—
Cipher with all white	7.9973	−0.0042	0.0187	−0.0021

**Table 8 entropy-22-00171-t008:** Comparative analysis.

Algorithm	Key space	Information entropy	UACI (%)	NPCR (%)
Ours	$3.8\times 10^{109}$	7.9971	33.5211	99.6130
Ref. [5]	$1.2\times 10^{83}$	7.9951	33.4624	99.4890
Ref. [15]	$1.9\times 10^{126}$	7.9973	30.2375	99.5950
Ref. [16]	$1.6\times 10^{79}$	7.9964	33.4694	99.6105
Ref. [19]	$2.9\times 10^{138}$	7.9845	28.6679	99.6101
Ref. [27]	$6.5\times 10^{119}$	7.9970	33.3443	99.7643

**Table 9 entropy-22-00171-t009:** Comparative analysis of the correlation coefficients of adjacent pixels.

Algorithm	Vertical	Horizontal	Diagonal
Ours	0.0020	0.0105	0.0019
Ref. [5]	0.0298	−0.0359	0.0052
Ref. [15]	0.0021	0.0004	−0.0038
Ref. [16]	0.0054	−0.0011	−0.0038
Ref. [19] Ref. [27]	0.0001 −0.0331	−0.0011 0.0125	−0.0014 −0.0236

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
