# Peer review of "A New Image Encryption Algorithm Based on Composite Chaos and Hyperchaos Combined with DNA Coding"

_entropy, 2020, doi:10.3390/e22020171_

Round 1
Reviewer 1 Report
In this paper, an image encryption algorithm is presented by the combination of a chaotic algorithm and a DNA type algorithm. It is well presented the current state, the proposed encryption algorithm and the chaotic algorithms, respectively the DNA on which it was based and also an analysis of the proposed algorithm was made. The paper is carefully writed, following all the writing stages.
At this paper I recommend to add the following elements:
- An analysis regarding the numerical precision knowing that the chaotic algorithms are very sensitive to the numerical precision;
- A comparative presentation with good well-known encryption algorithms regarding the key space and the statistical analysis
- A clear applicability of this algorithm, the operational scenario in which it can be applied. What would be the strengths in the real environment compared to the well-known algorithms used today in this field.
Author Response
Thank you very much for your comments, I will do my best to modify it. I will respond to your amendments one by one. The following is my response.
1. Question : An analysis regarding the numerical precision knowing that the chaotic algorithms are very sensitive to the numerical precision;
Answer : As far as I understand the analysis of numerical precision is the analysis of key sensitivity. This article explained in section 4.1.2. A small change to the system parameters, the results will be greatly different, as shown in Figure 7. It can be considered that the algorithm in this paper is highly sensitive to the initial value.
2. Question : A comparative presentation with good well-known encryption algorithms regarding the key space and the statistical analysis
Answer : Regarding the comparison of statistical analysis, this article analyzes the correlation between adjacent pixels. Analysis is added in lines 309 to 313 in Section 4.7. The comparison results are shown in Table 9.
3. Question : A clear applicability of this algorithm, the operational scenario in which it can be applied. What would be the strengths in the real environment compared to the well-known algorithms used today in this field.
Answer : This paper proposes a new one-dimensional composite system, which means that it has the advantages of one-dimensional chaos. In the actual environment, it has a simple structure and is easy to implement.

Reviewer 2 Report
This paper presents a new composite and Hyper-chaos based image encryption scheme. Generally, the paper is well written. However, I have some suggestions.
Please discuss some numeric values in abstract. For some statements, authors need references, for example,'Due to the high redundancy and correlation between image pixels'. Please provide solid references such as; https://doi.org/10.1007/s11071-015-2281-0 and https://link.springer.com/article/10.1007/s11042-015-2973-y Contribution of the work must be highlighted. In Fig. 5, after encryption why the last three steps such as DNA decoding etc are applied. Please change camera to cameraman in the draft paper. The key space of the algorithm in this paper reaches 4 × 10^128. How? Please explain and compare the keyspace with 10.3233/JIFS-17656. For histogram define x-axis and y-axis. Table 6 is confusing. Please plot all data on same page. Discuss computational complexity.Author Response
Thank you very much for your suggestion, I try my best to modify it, please forgive me if something is wrong. I divided your suggestion into 7 points and responded to you one by one. Here is my response.
1. Question : Please discuss some numeric values in abstract. For some statements, authors need references, for exampleï¼›Due to the high redundancy and correlation between image pixels'. Please provide solid references such as; https://doi.org/10.1007/s11071-015-2281-0 and https://link.springer.com/article/10.1007/s11042-015-2973-y Contribution of the work must be highlighted.
Answer : Analysis of experimental conclusions has been added to lines 20-24 in Abstract. And this paper added references in related descriptions. For example, the first reference is cited as Ref [1] on line 34, and the second reference is referred to as Ref [24] on line 223.
2. Question : In Fig. 5, after encryption why the last three steps such as DNA decoding etc are applied.
Answer : Figure 5 is a flowchart of the algorithm in this paper. The upper part is the encryption process, and the lower part is the decryption part. I'm sorry the previous description is unclear, and correct it in Figure 5. It is described in lines 186-188.
3. Question : Please change camera to cameraman in the draft paper.
Answer : Sorry for confusing the picture name. Corrected cameraman at line 190, and made corrections in Figures 6, 8, 9, and Tables 3, 4, 5.
4. Question : The key space of the algorithm in this paper reaches 4 × 10^128. How? Please explain and compare the key space with 10.3233/JIFS-17656.
Answer : My previous key space algorithm was wrong and corrected this. According to the international standard IEEE 754, the index part is expressed as a positive value to simplify the comparison. The double-precision floating-point type has 52 significant digits. And it is introduced in detail in section 4.1. Comparing the third reference, Ref [25], with this article, the result is listed in the second column of Table 8.
5. Question : For histogram define x-axis and y-axis.
Answer : Added supplementary notes on lines 223-226. The gray level histogram represents each gray level and the number of times that gray level occurs. The x-axis represents 0-255 grayscale values, and the y-axis represents the number of pixels in the corresponding grayscale in the figure.
6. Question : Table 6 is confusing.
Answer : Table 6 is PSNR with different noises and intensities. Added in lines 278 to 284. First increase the noise of the density of 0.001, 0.005 and 0.01 to the cipher images. The noised cipher images are shown in the first column of Table 6, and then they can be decrypted. The decrypted images are shown in the third column of Figure 6. The corresponding PSNR is shown in the fourth column. It can be seen from the figure that in the case of noise, the algorithm in this paper can decrypt the noised cipher images and obtain the original image information.
7. Question : Please plot all data on same page. Discuss computational complexity
Answer : I think the computational complexity is the time that it takes to record the encryption and decryption of different image sizes. Section 4.9 is added to supplement the text, and the results are shown in Figure 12.

Reviewer 3 Report
In this paper, a new image encryption scheme based on composite chaos and Qi hyperchaos combined with DNA coding is proposed. The paper is interesting but has some serious drawbacks:
Original images and description images should be compared with use of metrics. For example, SSIM. It 'll be better to add this comparison to Table 8. Part 4.1.1 should be expanded and better explained. Part 4.2.3 should be expanded and better explained. It seems to me that Information entropy is not so important for the paper. In order to test the anti-noise ability of the algorithm, this paper will add different intensity of Salt and Pepper noise to the ciphertext image and decrypt it. It's only for impulse noise. It'll be better to test the anti-noise ability of the algorithm by using different types of noise. The paper needs part Discussion.Author Response
Thank you very much for your suggestion, I try my best to modify it, please forgive me if something is wrong. I divided your suggestion into 4 points and responded to you one by one. Here is my response.
1. Question : Original images and description images should be compared with use of metrics. For example, SSIM. It 'll be better to add this comparison to Table 8.
Answer : I newly added the structural similarity in Section 4.8. It can be seen that in the range of 0 to 1, the higher the value, the less distortion the algorithm has. Please forgive me for my limited understanding. The SSIM value of the algorithm in this article is only found in section 4.8, and it is not listed in the table
2. Question : Part 4.1.1 should be expanded and better explained.
Answer : My previous key space algorithm was wrong and corrected this. According to the international standard IEEE 754, the index part is expressed as a positive value to simplify the comparison. The double-precision floating-point type has 52 significant digits. And it is introduced in detail in section 4.1.
3. Question : Part 4.2.3 should be expanded and better explained. It seems to me that Information entropy is not so important for the paper.
Answer : I reinterpreted it on lines 253-258. I think information entropy is also a common indicator of image quality evaluation. It reflects the richness of image information from the perspective of information theory. Generally, the larger the information entropy of an image, the richer its information content and the better its quality.
4. Question : In order to test the anti-noise ability of the algorithm, this paper will add different intensity of Salt and Pepper noise to the ciphertext image and decrypt it. It's only for impulse noise. It'll be better to test the anti-noise ability of the algorithm by using different types of noise. The paper needs part Discussion.
Answer : The two most commonly used noises are selected in Section 4.4 Anti-Noise Attack: Gaussian and Salt and Pepper Noise, and Table 6 has been updated.

Round 2
Reviewer 2 Report
Paper is revised significantly. I recommend this paper for publication.
Author Response
Thank you very much for your suggestions and comments on this article. I have learned a lot and thank you again for your work.
Reviewer 3 Report
The authors addressed almost all my remarks. It'll be better to add part Discussion.
Author Response
I'm sorry I didn't take a closer look at your comments so much that something was missing.
Question:The authors addressed almost all my remarks. It'll be better to add part Discussion.
Answer: I've added discussions on lines 327 to 336, and I've named this section Discussion in Section 5.
